# NGS Analysis Revealed Digenic Heterozygous *GCK* and *HNF1A* Variants in a Child with Mild Hyperglycemia: A Case Report

**DOI:** 10.3390/diagnostics11071164

**Published:** 2021-06-25

**Authors:** Fernanda Iafusco, Giovanna Maione, Cristina Mazzaccara, Francesca Di Candia, Enza Mozzillo, Adriana Franzese, Nadia Tinto

**Affiliations:** 1Department of Molecular Medicine and Medical Biotechnology, University of Naples “Federico II”, 80131 Naples, Italy; iafusco@ceinge.unina.it (F.I.); maione@ceinge.unina.it (G.M.); cristina.mazzaccara@unina.it (C.M.); 2CEINGE Advanced Biotechnology, 80131 Naples, Italy; 3Regional Center of Pediatric Diabetology, Section of Pediatrics, Department of Translational Medical Sciences, University of Naples “Federico II”, 80131 Naples, Italy; francesca.dicandia@unina.it (F.D.C.); mozzilloenza@gmail.com (E.M.); adriana.franzese@unina.it (A.F.)

**Keywords:** hyperglycemia, digenic variants, GCK/MODY, HNF1A/MODY

## Abstract

Monogenic diabetes (MD) represents a heterogeneous group of disorders whose most frequent form is maturity-onset diabetes of the young (MODY). MD is predominantly caused by a mutation in a single gene. We report a case of a female patient with suspected MD and a positive family history for diabetes and obesity. In this patient, two gene variants have been identified by next-generation sequencing (NGS): one in the Glucokinase (*GCK)* gene reported in the Human Gene Mutation Database (HGMD) and in the literature associated with GCK/MODY, and the other in the hepatocyte nuclear factor 1A (*HNF1A*) gene not previously described. The *GCK* variant was also identified in the hyperglycemic father, whereas the *HNF1A* variant was present in the mother. This new case of digenic *GCK/HNF1A* variants identified in a hyperglycemic subject, evidences the importance of NGS analysis in patients with suspected MD. In fact, this methodology will allow us to both increase the number of diagnoses and to identify mutations in more than one gene, with a better understanding of the genetic cause, and the clinical course, of the disease.

## 1. Introduction

Monogenic diabetes (MD) accounts for 1–6% of all cases of diabetes in patients of pediatric-adolescent age, but its prevalence is probably underestimated [1,2]. The most common type of MD is maturity-onset diabetes of the young (MODY), a clinically and genetically heterogeneous group of monogenic, non-insulin-dependent, and nonketotic diabetes mellitus characterized by early onset (commonly before age 25), the absence of autoimmunity, and beta-cell dysfunction [3].

Up to now, mutations in 14 genes involved in glucose homeostasis and pancreatic development have been associated with this form of diabetes [4]. Although it is an autosomal dominant disorder, de novo mutations should be considered in patients without a family history of diabetes [5]. The most common forms of MODY involve mutations in hepatocyte nuclear factor 1A (*HNF1A*), and glucokinase (*GCK*) genes, representing ~70% of all MODY subtypes [6,7]. The prevalence of GCK/MODY is higher in some countries (United States, Germany, Italy, France, and Spain) [8,9,10,11]; this may be because this form of diabetes is more easily identified in childhood in these countries.

Clinical manifestations of GCK/MODY are characterized by a slight increase in blood fasting glucose (5.5–8.0 mmol/L; 99–144 mg/dL), which is often present at birth [12]. Affected individuals are usually asymptomatic and the hyperglycemia is often discovered during routine medical examinations, such as during pregnancy or family screening when MODY is suspected. Micro- or macrovascular complications are extremely rare and no pharmacological treatment is recommended, but diet and regular physical activity are sufficient to maintain good glycemic control. Only in pregnancy may there be a need for treatment with insulin [13].

HNF1A/MODY is characterized by a progressive reduction in insulin secretion and, consequently, hyperglycemia. This requires treatment with drugs, and is associated with a tendency to develop microvascular complications [5]. The onset of the disease usually occurs during adolescence with an initial post-prandial hyperglycemia, which is then generalized in the patient.

Recognition of GCK/ and HNF1A/MODY, particularly when the patient is of pediatric age, is of fundamental importance, because this allows us to select the appropriate clinical management and to conserve healthcare resources. In fact, diabetic patients with HNF1A/MODY can avoid insulin and be treated with oral hypoglycemic agents such as sulfonylureas, to which they are particularly sensitive, whereas GCK/MODY patients, due to their mild phenotype, require less frequent clinical monitoring than patients with other forms of diabetes [14,15]. Therefore, molecular diagnosis of MODY may provide a perceptible impact on both the patient’s quality of life and the healthcare costs [16,17].

Herein, we report a new case of a digenic variant, identified by next-generation sequence analysis (NGS), in a hyperglycemic pediatric patient with suspected MD.

## 2. Case Presentation

We report the case of a female patient, the only child of non-consanguineous parents. The family history showed fasting hyperglycemia (135 mg/dL-7.5 mmol/L) and elevated hemoglobin-glycosylated HbA1C (6.3%-45 mmol/mol) in the father, and similar values in the paternal grandmother. In both cases, the condition was not treated. In the maternal family, the great-grandparents had a history of type 2 diabetes mellitus (T2D). The father and mother were both obese, with a body mass index (BMI) of 36 and 39, respectively. The mother and the mother’s sisters had thrombophilia (Figure 1A).

The patient came to our attention at 7 years of age, as a result of several episodes of fasting hyperglycemia (maximum glucose value detected 134 mg/dL-7.44 mmol/L) occasionally detected with a glucometer. Her medical history was characterized by birth at full-term from normal pregnancy, with her weight appropriate for gestational age (AGA). She did not present episodes of hypoglycemia at birth but she showed moderate to severe pulmonary valve stenosis, which was successfully treated with percutaneous valvuloplasty, and a speech delay that was diagnosed at 3 years of age.

Laboratory investigations showed glycemia 105 mg/dL (5.8 mmol/L), HbA1c 6.3% (45 mmol/mol). Her autoimmune diabetes autoantibodies (GAD, IAA, IA2, and ZnT8) were negative and her thyroid hormones (FT3, FT4, TSH) with autoantibodies (anti-tyreoglobulin, anti-peroxidase) were normal. An oral glucose tolerance test (OGTT) showed impaired glucose tolerance (IGT) without insulin resistance, and her liver and renal function tests were normal, with an adequate lipid profile: total cholesterol 167 mg/dL (4.32 mmol/L), LDL cholesterol 111 mg/dL (2.87 mmol/L), HDL cholesterol 50 mg/dL (1.29 mmol/L), triglycerides 91 mg/dL (1.03 mmol/L). Her BMI was 22.7 kg/m^2^ (BMIz-score: 2; >97th percentile), according to the curves of the World Health Organization (WHO), adjusted for age and sex. The patient was not undergoing any drug therapy.

Based on the early onset of the diabetes, the absence of beta-cell autoantibodies, and on the family history of diabetes, genetic tests for MD were performed.

## 3. Diagnostic Assessment

Peripheral blood samples for DNA analysis were collected after obtaining written informed consent from both the patient and her parents. Genomic DNA was extracted from leukocytes with a QIAamp DNA Blood Mini Kit (Qiagen, Hilden, Germany), according to the manufacturer’s instructions, and quantified by a NanoDrop ND-1000 spectrophotometer (ThermoFisher Scientific, Waltham, MA, USA). A next-generation sequencing (NGS) analysis, including 42 genes associated with non-autoimmune diabetes, was performed (Table 1). For each gene, we analyzed the coding regions, 50 bp in each of the intronic boundaries, the promoter, and the 3′UTR, for a total target size of about 1 Mb. A total of 50 ng of gDNA was processed through the SureSelectQXT Target Enrichment system (Agilent Technologies, Santa Clara, CA, USA) for Illumina multiplexed sequencing. Sequencing reactions were carried out on the MiSeq instrument (Illumina, San Diego, CA, USA). The sequence reads were aligned to the human reference genome (hg38) using the Alissa Align & Call v1.0.2.10 tool (Agilent Technologies, Santa Clara, CA, USA). The evidenced sequence variants were evaluated by Alissa Interpret v5.2.6 CE IVD software (Agilent Technologies, Santa Clara, CA, USA), using GRCh38.p2 and several databases of genomic variants [18].

The Alamut software ((http://www.interactive-biosofware.com/alamut-visual/) Alamut Visual v2.11.0, accessed on 23 March 2021)) was used to determine the pathogenicity of all missense variants. Bioinformatic tools available online, such as Sorting Intolerant From Tolerant (SIFT) (http://sift.jcvi.org/, accessed on 8 March 2021) and Polymorphism Phenotyping v2 (PolyPhen-2) (http://genetics.bwh.harvard.edu/pph2/, accessed on 8 March 2021) were utilized to classify identified variants. Further predictions were assessed with the Mutation Taster tool (http://www.mutationtaster.org, accessed on 8 March 2021) [19] and other tools included on the VarSome website (https://varsome.com/variant/hg38, accessed on 8 March 2021) [20,21]. All software programs were used with their default parameters. Finally, we classified the variants as pathogenic, likely pathogenic, variants of uncertain significance (VOUS), likely benign, or benign, in accordance with the standards and guidelines of the American College of Medical Genetics and Genomics (ACMG) [22]. 

Two nucleotide variants in exon 8 (c.868 G > T; p.Glu290*) of the *GCK* gene and in exon 4 (c.872 C > G; p.Pro291Arg) of the *HNF1A* gene were identified. These variants were confirmed with standard Sanger sequencing. Molecular sequencing extended to the diabetic parents showed that the *GCK* variant was present in the father and the *HNF1A* variant was present in the mother (Figure 1B).

## 4. Discussion

Monogenic diabetes comprises a group of heterogeneous genetic disorders characterized by the early onset of diabetes, the absence of autoimmunity, and beta-cell dysfunction [8,23].

Recognition of these forms of diabetes is crucial for reducing both the complications and the treatment costs associated with the disease, and to improve glycemic control with the most appropriate treatment and follow-up for patients [17,24]. In particular, HNF1A/MODY patients are generally responsive to sulfonylureas, whereas, for GCK/MODY patients, no pharmacological treatment is recommended and diet and regular physical activity are sufficient to maintain good glycemic control [6,14].

In this case report, we described an unusual molecular diagnosis performed by NGS analysis in a hyperglycemic child, which revealed two distinct genetic variants in the *GCK* and *HNF1A* genes. The *GCK* variant, reported in the Human Gene Mutation Database (HGMD) (http://www.hgmd.cf.ac.uk.ac, accessed on 8 March 2021) and in the literature as associated with GCK/MODY [12], was also identified in the diabetic father. The patient presented a moderate hyperglycemia consistent with a GCK/MODY diagnosis.

The *HNF1A* variant, not reported in HGMD or in any known genomic variants database, has been classified as novel and its pathogenicity has been evaluated. In particular, according to ACMG classification criteria [21], we added fundamental parameters other than those by default in the VarSome database and therefore classified p.Pro291Arg as “likely pathogenic” because it has a low frequency (combined minor allele frequency, MAF < 1%), it is located in a mutational hotspot (PM1), it is a missense variant in a gene that has a low rate of benign missense variation (PP2), other different changes in 291 amino acid residues have been described as pathogenic (PM5), and a reputable source (NCBI ClinVar) (https://www.ncbi.nlm.nih.gov/, accessed on 8 March 2021) reports that the variant is likely pathogenic (PP5). Moreover, the variant is located in a conserved region among species, indicating its functional relevance, and wild-type amino acid Proline has physicochemical differences with respect to Arginine. Furthermore, because computational prediction tools were not all in accord, they were not considered as criteria for classifying the variant.

The p.Pro291Arg is also present in the mother (40 years) who has not shown glycemic alterations until now.

It is known that the clinical expression of HNF1A/MODY is highly variable, even within the same family [3,25]. In fact, *HNF1A* mutation carriers may be normoglycemic while their mutated siblings may be hyperglycemic, as reported by Miedzybrodzka et al. [26]. The authors described a family with a missense mutation in the *HN1A* gene that was not fully penetrant, because it was present in all diabetic family members even if with considerable variations in severity and age of diagnosis (ranging from 16 to 67 years), and in two non-diabetic family members (aged 87 and 46) [26]. This case confirms that, among the *HNF1A* variants, some are sufficient for diabetes onset; however, not all carriers of these variants have early-onset diabetes and some do not have diabetes at all. It was hypothesized that multiple loci, inherited independently of diabetes, are modifiers of the HNF1A/MODY phenotype [25,27]. Some of these loci may exert their effect at the level of the beta-cell, enhancing the insulin secretion deficit caused by *HNF1A* mutations; others may influence the onset of diabetes by affecting the amount of insulin that peripheral tissues require to maintain glycemia at the normal level [25].

This phenotypic variability and incomplete penetrance have been also described in HNF4A/MODY [28,29] and were found in our clinical experience too (unpublished data). 

As HNF1A/MODY is usually onset at a post-pubertal age, we could hypothesize that our patient, thus far showing a mild phenotype compatible with GCK/MODY, due to the progressive beta-cell function failure caused by the *HNF1A* mutation, will develop a more severe phenotype later in life. However, studies on phenotypic effects in the general population of rare variants in MODY genes, previously reported as pathogenic, have provided evidence that the majority of variant carriers remain euglycemic through middle age, highlighting the limitations of disease variant databases, as well as bioinformatics criteria, in assigning pathogenicity to rare variants [30]. Genetic and functional studies have also demonstrated that missense variants in *HNF1A* can have a spectrum of effects, ranging from weak effects, predisposing patients to T2D, to highly penetrant effects, causing MODY [30,31]. 

Few reports are present in the literature describing individuals with digenic variants in different MODY genes [32,33,34,35,36]. One of these reported, for the first time, the co-inheritance of *HNF1A* and *GCK* mutations in two patients of the same family with the coexistence of two phenotypes, GCK/MODY and HNF1A/MODY [35]. In particular, the adult patient showed typical HNF1A/MODY characteristics and the presence of the second GCK mutation did not modify the diabetic phenotype, whereas the younger patient with the same genotype showed, as did our patient, only impaired fasting glucose (IFG), which was likely due to a shorter disease evolution. 

This new case of digenic *GCK/HNF1A* variants, evidenced in a hyperglycemic pediatric patient by NGS analysis, confirms that the eventuality of two mutations in different MODY genes should not be excluded in patients with suspect of MD. In particular, in our proband, because we cannot exclude a possible evolution of the disease, a more careful clinical follow-up is necessary for accurate management and treatment. In addition, considering the relevance of *HNF1A* variants in the contribution of T2D risk, a greater attention to life style and a correction of the weight is strongly suggested for the mother carrier of the novel *HNF1A* variant.

## Figures and Tables

**Figure 1 diagnostics-11-01164-f001:**
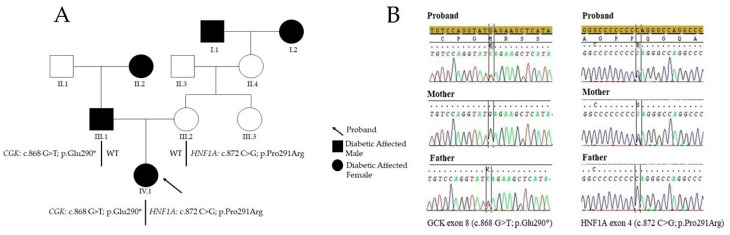
(**A**) Pedigree of the analyzed family. The pedigree shows the segregation of the variants detected in the proband and her family. In the subject III.1, the variant, carried in the heterozygous status, is the c.868 G > T; p.Glu290*, in the glucokinase (CGK) gene; the III.2 subject carried the c.872 C > G; p.Pro291Arg, in the HNF1A gene. The proband of the study (IV.1) shows all the two variants cited above in the parents. I.1 and I.2 proband’s maternal great-grandparents: diabetes; II.2 proband’s paternal grandmother: diabetes; III.1 proband’s father: diabetes, obesity; III.2 proband’s mother: obesity, thrombophilia; III.3 proband’s aunt: thrombophilia; IV.1: diabetes (**B**) GCK and HNF1A Sanger sequences of the proband, the mother, and the father.

**Table 1 diagnostics-11-01164-t001:** Panel of genes associated with monogenic diabetes, analysed in the proband.

NGS GENE PANEL						
**ABCC8**	**CEL**	**GCK**	**HNF4A**	**KLF11**	**PDX1**	**SLC16A1**
AKT2	CISD2	GLIS3	IER3IP1	LMNA	PLIN1	SLC19A2
APPL1	EIF2AK3	GLUD1	INS	MAFA	POLD1	SLC9B1
ASB14	FOXP3	HADH	INS-IGF2	NEUROD1	PPARG	TRMT10A
BLK	GATA4	HNF1A	INSR	NEUROG3	PTF1A	WFS1
C12orf43	GATA6	HNF1B	KCNJ11	PAX4	RFX6	ZFP57

## Data Availability

The authors confirm that the data supporting the findings of this study are available within the article can be made available upon reasonable request.

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
