# Peer review of "NGS Analysis Revealed Digenic Heterozygous GCK and HNF1A Variants in a Child with Mild Hyperglycemia: A Case Report"

_diagnostics, 2021, doi:10.3390/diagnostics11071164_

Round 1

Reviewer 1 Report

Dear Editor,

I carefully read the manuscript by Iafusco et al, that is interesting and well written. I suggest the authors to include a genealogical tree in the manuscript. Other than fasting plasma glucose and HbA1c, I suggest the authors to include in the manuscript also other metabolic parameters that are often impaired in diabetes (e.g. total cholesterol, triglycerides, HDL-cholesterol and thyroid function), if available.

Author Response

Ref.: diagnostics 1257285

Fernanda Iafusco, Giovanna Maione, Cristina Mazzaccara, Francesca Di Candia, Enza Mozzillo, Adriana Franzese and Nadia Tinto

NGS Analysis Revealed Digenic Heterozygous GCK and HNF1A Variants in a Child with Mild Hyperglycemia: A Case Report

Reviewer 1

I carefully read the manuscript by Iafusco et al, that is interesting and well written. I suggest the authors to include a genealogical tree in the manuscript. Other than fasting plasma glucose and HbA1c, I suggest the authors to include in the manuscript also other metabolic parameters that are often impaired in diabetes (e.g. total cholesterol, triglycerides, HDL-cholesterol and thyroid function), if available.

We thank the reviewer for the comments. As suggested we added the genealogical tree in the Fig.1 panel A and included in the text pag 2, line 74.

Lipid profile and thyroid function of the proband and her parents were normal as just previously reported in the text. Nevertheless, as suggested by the reviewer, we included total cholesterol, LDL-cholesterol, HDL-cholesterol and triglycerides only for the proband. Text  pag 2, line 87-89.

Reviewer 2 Report

The case described by Iafusco et.al. is an interesting clinical case as it highlights the digenic GCK/HNF1A variant in a MODY subject identified using a next-generation sequencing approach. Both the mutations were also detected in either of the parents suggesting familial inheritance and link with diabetic history in the family. Tracing these mutations in the parental pedigree would have provided critical information about the genetic information of these mutations. Nonetheless, the case study is well-conducted and presented. There was no significant plagiarism reported in the case study. The case study should be accepted in the current form.  

Author Response

Ref.: diagnostics 1257285

Fernanda Iafusco, Giovanna Maione, Cristina Mazzaccara, Francesca Di Candia, Enza Mozzillo, Adriana Franzese and Nadia Tinto

NGS Analysis Revealed Digenic Heterozygous GCK and HNF1A Variants in a Child with Mild Hyperglycemia: A Case Report

Reviewer 2

The case described by Iafusco et.al. is an interesting clinical case as it highlights the digenic GCK/HNF1A variant in a MODY subject identified using a next-generation sequencing approach. Both the mutations were also detected in either of the parents suggesting familial inheritance and link with diabetic history in the family. Tracing these mutations in the parental pedigree would have provided critical information about the genetic information of these mutations. Nonetheless, the case study is well-conducted and presented. There was no significant plagiarism reported in the case study. The case study should be accepted in the current form. 

We want to thank the reviewer for the comment.